# Efficient Propylene/Ethylene Separation in Highly Porous Metal–Organic Frameworks

**DOI:** 10.3390/ma16010154

**Published:** 2022-12-23

**Authors:** Xiao-Min Liu, Lin-Hua Xie, Yufeng Wu

**Affiliations:** 1Institute of Circular Economy, Beijing University of Technology, Beijing 100124, China; 2Beijing Key Laboratory for Green Catalysis and Separation and Department of Environmental Chemical Engineering, Faculty of Environment and Life, Beijing University of Technology, Beijing 100124, China

**Keywords:** metal–organic framework, light hydrocarbon separation, ethylene, propylene, activated carbon

## Abstract

Light olefins are important raw materials in the petrochemical industry for the production of many chemical products. In the past few years, remarkable progress has been made in the synthesis of light olefins (C2–C4) from methanol or syngas. The separation of light olefins by porous materials is, therefore, an intriguing research topic. In this work, single-component ethylene (C_2_H_4_) and propylene (C_3_H_6_) gas adsorption and binary C_3_H_6_/C_2_H_4_ (1:9) gas breakthrough experiments have been performed for three highly porous isostructural metal–organic frameworks (MOFs) denoted as Fe_2_M-L (M = Mn^2+^, Co^2+^, or Ni^2+^), three representative MOFs, namely ZIF-8 (also known as MAF-4), MIL-101(Cr), and HKUST-1, as well as an activated carbon (activated coconut charcoal, SUPELCO^©^). Single-component gas adsorption studies reveal that Fe_2_M-L, HKUST-1, and activated carbon show much higher C_3_H_6_ adsorption capacities than MIL-101(Cr) and ZIF-8, HKUST-1 and activated carbon have relatively high C_3_H_6_/C_2_H_4_ adsorption selectivity, and the C_2_H_4_ and C_3_H_6_ adsorption heats of Fe_2_Mn-L, MIL-101(Cr), and ZIF-8 are relatively low. Binary gas breakthrough experiments indicate all the adsorbents selectively adsorb C_3_H_6_ from C_3_H_6_/C_2_H_4_ mixture to produce purified C_2_H_4_, and 842, 515, 504, 271, and 181 cm^3^ g^−1^ C_2_H_4_ could be obtained for each breakthrough tests for HKUST-1, activated carbon, Fe_2_Mn-L, MIL-101(Cr), and ZIF-8, respectively. It is worth noting that C_3_H_6_ and C_2_H_4_ desorption dynamics of Fe_2_Mn-L are clearly faster than that of HKUST-1 or activated carbon, suggesting that Fe_2_M-L are promising adsorbents for C_3_H_6_/C_2_H_4_ separation with low energy penalty in regeneration.

## 1. Introduction

Light olefins, particularly ethylene (C_2_H_4_) and propylene (C_3_H_6_), are important raw materials in the petrochemical industry for the manufacture of products such as plastics, solvents, cosmetics, paints, and drugs. Light olefins are traditionally produced from the thermal or catalytic cracking of crude oil. In the past decades, considerable attention has been paid to the production of light olefins (C_2_^=^–C_4_^=^) from other alternative feedstocks, such as coal, natural gas, and biomass, by the Fischer–Tropsch synthesis (FTS) and the methanol-to-olefins (MTO) reaction [1,2]. Some recent breakthrough results showed that coal- and biomass-derived syngas (a mixture of carbon monoxide and hydrogen) could be converted to light olefins in very high selectivities (up to 80%, relative to less valuable saturated hydrocarbons) by advanced catalysts [3,4]. These new innovations would lead to a high demand for efficient separation technology for light olefins in the future. Nowadays, the separation of C_2_H_4_ and C_3_H_6_ is commonly accomplished by cryogenic distillation, which is a mature but energy-intensive process. Many works have been reported to develop new materials and technologies to efficiently separate C_2_H_4_ and C_3_H_6_ [5,6,7,8,9]. Adsorptive separation of C_2_H_4_ and C_3_H_6_ with porous materials is regarded as a promising alternative [10].

Metal–organic frameworks (MOFs), a type of porous material composed of metal ions/clusters and organic ligands, have attracted intensive research attention in the past two decades [11,12,13,14]. The high porosity, tailorable structure, crystalline nature, and facile synthesis of MOFs endow this type of material with great potential in gas storage, separation, catalysis, and sensing, among others. Particularly, some recent works have shown that bimetallic MOFs could exhibit improved performance in gas adsorption and catalysis [15,16,17,18]. Separation is one of the most studied applications where the intrinsic porous structures of MOFs could be utilized. It has been demonstrated that MOFs are high potential in light hydrocarbon separations [19,20,21,22,23,24,25,26,27,28,29,30,31,32,33,34,35,36,37]. Relatively, the separation of C_2_H_4_ and C_3_H_6_ using MOFs is less explored, although C_2_H_4_ and C_3_H_6_ adsorption isotherms of some MOFs were reported [38,39,40].

In this work, the C_3_H_6_/C_2_H_4_ separation performances of three new isostructural MOFs, [Fe_2_M(*µ*_3_-O)(L)_2_] (denoted as Fe_2_M-L, M = Mn^2+^, Co^2+^, or Ni^2+^, H_3_L = [1,1′:3′,1″-terphenyl]-4,4″,5′-tricarboxylic acid), three representative MOFs, ZIF-8 (a zinc(II) 2-methylimidazolate also known as MAF-4) [41,42], MIL-101(Cr) (a chromium(III) terephthalate) [43], and HKUST-1 (a copper(II) benzene-1,3,5-tricarboxylate) [44], and a commercial activated carbon (activated coconut charcoal, SUPELCO^©^) have been evaluated by single-component gas adsorption isotherm measurements and binary gas mixture breakthrough experiments. Their C_3_H_6_ and C_2_H_4_ adsorption properties and C_3_H_6_/C_2_H_4_ separation performances have been compared and discussed.

## 2. Materials and Methods

### 2.1. Materials and Instrumentation

All the chemicals were purchased from chemical suppliers and utilized without purification. ZIF-8, MIL-101(Cr), and HKUST-1 were prepared by previously reported methods [45,46,47]. Activated carbon (activated coconut charcoal, SUPELCO^©^ (Bellefonte, PA, USA) was purchased from Sigma-Aldrich (Missouri, USA). Thermal gravimetric analyses (TGA) were carried out under zero air conditions with a SHIMADZU TGA-50 thermogravimetric analyzer (heating rate: 5 °C/min) (Kyoto, Japan). Fourier transform infrared (FT-IR) spectra were measured with a Shimadzu IRAffinity FT-IR spectrophotometer (Kyoto, Japan). The adsorption isotherms of N_2_ and hydrocarbons were recorded with an ASAP2020 adsorption analyzer (Micromeritics, Norcross, GA, USA). PXRD measurements were performed with a Smartlab3 X-ray powder diffractometer (Rigaku, Tokyo, Japan). Inductively coupled plasma-atomic emission spectroscopy (ICP-AES) measurements were performed with a PerkinElmer Optima 8300 spectrometer (Waltham, MA, USA). 

### 2.2. Synthesis

Syntheses of [Fe_2_M(*µ*_3_-O)(CH_3_COO)_6_] (the precursors of Fe_2_M-L, M = Mn^2+^, Co^2+^, or Ni^2+^): an aqueous solution (70 ml) of Na(CH₃COO) 3H₂O (42 g, 0.31 mol) was added to an aqueous solution (70 ml) of Fe(NO₃)₃·9H₂O (8 g, 0.02 mol) and M(NO₃) (0.1 mol). Dark-red precipitate was formed, which was collected after filtration, washing with solvents (water and ethanol), and drying in the air [17].

Syntheses of Fe_2_M-L: to a 5 mL glass vial, [Fe_2_M(*µ*_3_-O)(CH_3_COO)_6_] (~0.015 g, 0.05 mmol) and the ligand H_3_L (H_3_L = 1,1′:3′,1″-terphenyl]-4,4″,5′-tricarboxylic acid) (0.012 g, 0.05 mmol) were added. Then, the solvent N,N-dimethylformamide (DMF, 2 mL) and the coordination modulator acetic acid (0.25 mL) were introduced to the vial. The solids were all dissolved after an ultrasonication treatment, and the vial was capped and placed in a preheated oven (120 °C) for 72 h. Block-shaped dark yellow or brown single crystals were formed and collected (yield: ca. 81% based on Fe).

### 2.3. Single-Crystal X-ray Diffraction

Single crystals of Fe_2_M-L were picked for single-crystal diffraction experiments. The diffraction data were collected at 100 K with a Rigaku Supernova CCD diffractometer (Tokyo, Japan) equipped with a mirror-monochromatic enhanced Cu-*K*α radiation (*λ* = 1.54184 Å). The dataset was corrected by empirical absorption correction using spherical harmonics, implemented in the SCALE3 ABSPACK scaling algorithm. The structure was solved by direct methods and refined by full-matrix least-squares on *F*^2^ with anisotropic displacement using the SHELXTL software package (Göttingen, Germany) [48]. Non-hydrogen atoms on the frameworks were refined with anisotropic displacement parameters during the final cycles. The hydrogen atoms on the ligands were positioned geometrically and refined by using a riding model. The electron density of the disordered guest molecules in Fe_2_M-L was flattened by using the SQUEEZE routine of PLATON (Utrecht, The Netherlands) [49]. The graphical representations of single-crystal structures of Fe_2_M-L were performed by the Diamond software (Bonn, Germany) [50]. Some strong residual Q peaks were found near the water molecules coordinated with the metal ions in the final refinement cycles, indicating that partially coordinated solvent molecules may be DMF molecules rather than water molecules, which could not be fully modeled. The single-crystal structure data of Fe_2_M-L have been deposited in the Cambridge Crystallographic Data Centre (CCDC deposition number: 2177907–2177909).

### 2.4. Estimation of C_3_H_6_ and C_2_H_4_ Adsorption Heat

The C_3_H_6_ and C_2_H_4_ adsorption heats were calculated from the adsorption data recorded at 298 and 273 K. The Toth equation [51] (Equation (1)) was first utilized for fittings the adsorption isotherms (Appendix A), where *N* stands for the gas uptake, *N_sat_* stands for the saturated uptake, *P* represents pressure, *b* and *t* represent two constants.
(1)$$N=\frac{N_{sat}bP}{{\left(1+bP^{t}\right)}^{\frac{1}{t}}}$$
(2)$$P=\frac{N}{{\left(b^{t}N_{sat}{}^{t}-bN^{t}\right)}^{\frac{1}{t}}}$$

The following Equation (2) could be obtained from the rearrangement of Equation (1).

The C_3_H_6_ and C_2_H_4_ adsorption heats (*Q_st_*) were estimated by the Clausius–Clapeyron equation [52] (Equation (3)), where *C* represents a constant, *R* stands for the universal gas constant, and *T* stands for temperature. Assuming (ln*P*)*_N_* is the function, and (1/*T*) is the variable, *Q_st_* depending on the gas uptake (*N*), could be calculated from the slopes data points (*Q_st_*/*R*).
(3)$${\left(\ln P\right)}_{N}=-\frac{Q_{st}}{R}\frac{1}{T}+C$$

### 2.5. Prediction of IAST C_3_H_6_/C_2_H_4_ Selectivity

Ideal adsorbed solution theory [53] (IAST) is a well-accepted way to predict gas adsorption selectivity for gas mixtures by adsorption data of the individual gases. The IAST defines the following equations.
(4)$$y_{1}+y_{2}=1$$
(5)$$x_{1}+x_{2}=1$$
(6)$$p_{mix}y_{1}=p_{1}^{o}x_{1}$$
(7)$$p_{mix}y_{2}=p_{2}^{o}x_{2}$$
(8)$$\pi_{1}^{o}=\frac{RT}{A}\int_{0}^{p_{1}^{o}}n_{1}(p)d\ln p$$
(9)$$\pi_{2}^{o}=\frac{RT}{A}\int_{0}^{p_{2}^{o}}n_{2}(p)d\ln p$$
(10)$$\pi =\pi_{1}^{o}=\pi_{2}^{o}$$

In these equations, *R* stands for the universal gas constant, *T* stands for the temperature for adsorption experiment, *A* represents the surface area of adsorbent, *x_i_* stands for the ratio of gas *i* in the gas mixture adsorbed by adsorbent, *y_i_* stands for the ratio of gas *i* in the gas mixture before adsorption, *p*_mix_ represents the pressure of gas mixture before adsorption, *p_i_*^0^ represents the pressure of gas *i* that corresponds to the spreading pressure *π* of the binary mixture, *n_i_*(*p*) stands for the uptake of gas *i* at the pressure *p*.

From Equations (4)–(10), Equation (11) is obtained.
(11)$$\int_{0}^{\frac{p_{mix}y_{1}}{x_{1}}}n_{1}(p)d\ln p=\int_{0}^{\frac{p_{mix}y_{2}}{1-x_{1}}}n_{2}(p)d\ln p$$
(12)$$S_{12}=\frac{x_{1}y_{2}}{x_{2}y_{1}}$$

The adsorption selectivity of gas 1 over gas 2 (*S*_12_) can be obtained by Equation (12).

For the adsorption of 1:9 C_3_H_6_/C_2_H_4_ gas mixture at 1 bar, C_3_H_6_ is gas 1, C_2_H_4_ is gas 2, *p*_mix_ is 1 bar, *y*_1_ is 0.1, *y*_2_ is 0.9, and *n_i_*(*p*) can be calculated by fittings adsorption data of the individual gases (Appendix A). *x*_1_ can then be calculated by solving Equation (11). At last, *x*_2_ can be obtained by applying the *x*_1_ to Equation (5), and *S*_12_ can be calculated from Equation (12). The C_3_H_6_/C_2_H_4_ selectivities for a 1:1 C_3_H_6_/C_2_H_4_ gas mixture or at other pressures could be calculated similarly.

### 2.6. Gas Mixture Breakthrough

Breakthrough tests for the gas mixture were performed with a 1:9 C_3_H_6_/C_2_H_4_ gas mixture. Quartz tubes (6 mm for outer diameter, 3 mm for inner diameter, and 100 mm in length) were packed with the adsorbents. The adsorbents were first activated at 100 °C overnight inside the tubes under a He flow with a flow rate of 10 SCCM. Mass flow controllers (Alicat Scientific, Tucson, AZ, USA) were used to control the gas flow rate. After the adsorbents were cooled down to room temperature, the breakthrough experiments started when the He flow was changed into a flow of the C_3_H_6_/C_2_H_4_ gas mixture (flow rate: 2 SCCM). The concentrations of C_3_H_6_ and C_2_H_4_ gases at outlet were determined by a mass spectrometer (Hiden HPR20). For monitoring the desorption dynamics of the adsorbed C_3_H_6_ and C_2_H_4_ gases in the adsorbents, the gas flow was changed from the binary C_3_H_6_/C_2_H_4_ gas to a He flow of 10 SCCM.

## 3. Results and Discussion

### 3.1. Crystal Structure and Porosity

Fe_2_M-L were synthesized from solvothermal reactions of [Fe_2_M(*µ*_3_-O)(CH_3_COO)_6_] precursors and H_3_L ligand in DMF at 120 °C. It was found that using the pre-synthesized [Fe_2_M(*µ*_3_-O)(CH_3_COO)_6_] precursors instead of mixtures of Fe/M metal salts as metal ion source is necessary for the formation of final crystalline products. Additionally, the introduction of an excess coordination modulator (acetic acid) is important for the production of Fe_2_M-L crystals in high crystallinity and with a relatively large size (>0.1 mm). It is well-known that coordination modulators affect the nucleation and growth rate, morphology, and crystallinity of MOFs, and only intergrown aggregates with poor crystallinity or even amorphous phases could be obtained without modulators in some cases [17,54]. The presence of two types of metal ions with a Fe to M ratio of 2:1 in Fe_2_M-L was confirmed by ICP-AES measurements for digested samples of the MOFs (Appendix A).

Single-crystal X-ray diffraction (SCXRD) experiments and structure analyses revealed that Fe_2_M-L are isostructural and in the *R*-3*c* space group (trigonal) (Appendix A). There is one type of [Fe_2_M(*µ*_3_-O)(−COO)_6_] (denoted as Fe_2_M hereafter) clusters and one type of L^3−^ ligands in their structures. Each Fe_2_M cluster is connected with six different but equivalent L^3−^ ligands, and each L^3−^ ligand bridges three Fe_2_M clusters (Figure 1a). The interconnection of the Fe_2_M clusters and L^3−^ ligands in such a way results in their 3D frameworks, which could be regarded as (3,6)-connected nets from a topological point of view (point symbol: {4·6^2^}_2_{4^2^·6^7^ 8^6^}). The frameworks contain large volumes which are occupied by disordered guest molecules, as indicated by TGA results (Appendix A). The solvent-accessible volumes are ~74% of the whole structures, as estimated by Platon [49].

The highly open frameworks of Fe_2_M-L can also be regarded as alternate packing of two sets of cages. One set is octahedral, each of which consists of 6 Fe_2_M clusters and 8 L^3−^ ligands (Figure 1b). In the octahedral cage, two neighboring Fe_2_M clusters are all bridged by one L^3−^ ligand with its two carboxylate groups. The other set of cages is in a tetrakaidecahedron shape, each of which is built from 12 Fe_2_M clusters and 12 L^3−^ ligands, where each L^3−^ ligand links 3 neighboring Fe_2_M clusters (Figure 1c). The cavities inside the octahedral and tetrakaidecahedral cages are ~6 and ~9 Å, respectively. Each small cage is surrounded by eight large cages, and each large cage is surrounded by eight small cages and six equivalent large cages by polyhedral face sharing (Figure 1d). The windows between two small cages or between one large cage and one small cage are in a diameter of ~3.8 Å, and the windows between two large cages are in a diameter of ~5.1 Å. Several isoreticular MOFs to Fe_2_M-L were previously reported [55,56,57].

To confirm the permanent porosity of Fe_2_M-L, N_2_ adsorption isotherms were recorded at 77 K after the as-prepared crystals were activated by guest exchange (methanol as solvent) and then degassing at 80 °C. The three MOFs showed highly similar N_2_ adsorption isotherms (Figure 2a), which is consistent with their isostructural structures and close unit cell parameters. The isotherms are type I isotherms typical for microporous materials, showing saturated N_2_ adsorption capacities of 837, 854, and 847 cm^3^ g^−1^ at ~1 *P*/*P*_0_ for Fe_2_Mn-L, Fe_2_Co-L, and Fe_2_Ni-L, respectively. The apparent BET/Langmuir surface areas of the three MOFs are estimated to be 3105/3600, 3168/3675, and 3169/3674 m^2^ g^−1^, respectively. The pore volumes are estimated to be 1.29, 1.32, and 1.31 cm^3^ g^−1^, respectively, which are almost the same as the predicted pore volumes from their single crystal data (1.30, 1.32, and 1.32 cm^3^ g^−1^). The results suggested that the MOF samples were in a pure phase, and their highly open frameworks remained unchanged after the evacuation of guests. The high purity of the batch MOF crystal samples was also confirmed by PXRD measurement results, which showed a good agreement between the PXRD patterns of the MOF samples and the single-crystal structure simulated ones (Appendix A).

### 3.2. Adsorption Study for C_2_H_4_ and C_3_H_6_

The adsorption isotherms of C_2_H_4_ and C_3_H_6_ were recorded for Fe_2_M-L at 298 K as well as 273 K. As shown in Figure 2b, the three MOFs showed close gas adsorption capacities at all pressure ranges. The C_2_H_4_ uptakes were 87.5, 94.4, and 94.9 cm^3^ g^−1^, and the C_3_H_6_ uptakes are 291.1, 302.3, and 304.2 cm^3^ g^−1^ at 1 bar for Fe_2_Mn-L, Fe_2_Co-L, and Fe_2_Ni-L, respectively. The slightly lower gas uptakes of Fe_2_Mn-L with respect to those of the other two MOFs are consistent with the results of 77 K N_2_ adsorption studies, which revealed that Fe_2_Mn-L had a slightly lower porosity than Fe_2_Co-L and Fe_2_Ni-L. Overall, the results indicate the gas adsorption properties of Fe_2_M-L are not very dependent on the nature of the M ions. Therefore, only Fe_2_Mn-L was investigated for further experiments. For comparison, C_2_H_4_ and C_3_H_6_ adsorption measurements were also carried out for four benchmark adsorbents, namely, HKUST-1, MIL-101(Cr), ZIF-8, and a commercial activated carbon. ZIF-8, MIL-101(Cr), and HKUST-1 were all prepared by reported methods [45,46,47], and activated carbon was purchased from Sigma-Aldrich. Before the C_2_H_4_ and C_3_H_6_ adsorption measurements, PXRD measurements and/or N_2_ adsorption experiments at 77 K were performed for those adsorbents (Appendix A). According to the adsorption experiments, their porosities are evaluated and shown in Table 1.

As shown in Figure 2c, for HKUST-1 and activated carbon, the C_3_H_6_ uptakes increase abruptly at the low-pressure range (~140 and 90 cm^3^ g^−1^ at 0.1 bar), and after a gradual increase at the high-pressure range, the uptakes reach 167.0 and 128.9 cm^3^ g^−1^ at 1 bar, respectively. The C_2_H_4_ adsorption isotherms of the two adsorbents share a similar profile to the C_3_H_6_ adsorption isotherms with lower uptakes, being 136.5 and 98.8 cm^3^ g^−1^ at 1 bar, respectively. In contrast, for MIL-101(Cr) and ZIF-8, the uptakes of C_3_H_6_ or C_2_H_4_ all gradually increase in the full pressure range. Notably, although MIL-101(Cr) has a larger pore volume (1.47 cm^3^ g^−1^) than Fe_2_Mn-L (1.29 cm^3^ g^−1^), its C_3_H_6_ and C_2_H_4_ uptakes at 1 bar (196.6 and 62.1 cm^3^ g^−1^) is obviously lower than those of Fe_2_Mn-L (291.1 and 87.5 cm^3^ g^−1^). Additionally, the gas uptakes of ZIF-8 are the lowest among all the tested adsorbents, being 80.6 and 26.4 cm^3^ g^−1^ at 1 bar for C_3_H_6_ and C_2_H_4_, respectively, although its pore volume is clearly higher than those of HKUST-1 and activated carbon (Table 1). The C_3_H_6_ uptake of Fe_2_Mn-L at 1 bar is higher than those of the other adsorbents, but its C_2_H_4_ uptake is lower than that of HKUST-1 and activated carbon. The C_3_H_6_/C_2_H_4_ uptake ratios at 1 bar are 3.3, 3.2, 3.1, 1.3, and 1.2 for Fe_2_Mn-L, MIL-101(Cr), ZIF-8, activated carbon, and HKUST-1, respectively. The C_3_H_6_ and C_2_H_4_ adsorption isotherms of MIL-101(Cr), ZIF-8, and HKUST-1 were also previously reported [20,38,40,58,59,60,61], and those reported results are basically consistent with those presented in this work. Some slight differences may result from the subtle difference in sample preparation and/or activation. 

To assess their capability of selectively adsorbing C_3_H_6_ from C_3_H_6_/C_2_H_4_ mixture, the IAST selectivities [53] were predicted for the five adsorbents to 1:1 and 1:9 binary C_3_H_6_/C_2_H_4_ gases, respectively. The results show that C_3_H_6_/C_2_H_4_ selectivities for HKUST-1 and activated carbon are higher than those for Fe_2_Mn-L, MIL-101(Cr), and ZIF-8, especially at low-pressure range (Figure 3a,b). The selectivities of HKUST-1, activated carbon, Fe_2_Mn-L, ZIF-8, and MIL-101(Cr) at 1 bar are 16.3, 11.6, 7.8, 6.7, and 6.6 for the 1:1 binary C_3_H_6_/C_2_H_4_ gas, and 18.0, 14.4, 7.6, 7.0, and 6.3 for the binary 1:9 C_3_H_6_/C_2_H_4_ gas, respectively. Noteworthily, the order of the adsorbents in their IAST selectivities at 1 bar (HKUST-1 > activated carbon > Fe_2_Mn-L > ZIF-8 > MIL-101(Cr)) is dramatically different from the order of the adsorbents in their C_3_H_6_/C_2_H_4_ uptake ratios at 1 bar in adsorption experiments of pure gases (Fe_2_Mn-L > MIL-101(Cr) > ZIF-8 > activated carbon > HKUST-1).

For a better understanding of the C_3_H_6_ and C_2_H_4_ adsorption behavior and the IAST predicted C_3_H_6_/C_2_H_4_ selectivities, the C_3_H_6_ and C_2_H_4_ adsorption heats (*Q*_st_) were estimated for the adsorbents by Clausius–Clapeyron equation using Toth equation fitting parameters from adsorption results obtained at 273 and 298 K (Appendix A) [51,52]. For C_2_H_4_ adsorption, the *Q*_st_ values for HKUST-1, activated carbon, Fe_2_Mn-L, and MIL-101(Cr) decrease as the loadings increase, whereas the *Q*_st_ values for ZIF-8 increase at low loading and stabilize at higher loadings (Figure 3c). The *Q*_st_ values at low loading for HKUST-1, activated carbon, Fe_2_Mn-L, MIL-101(Cr) (45.1, 32.5, 38.9, 35.8 kJ mol^−1^) are obviously larger than the *Q*_st_ of ZIF-8 (13.8 kJ mol^−1^), which is similar to the vaporization enthalpy of C_2_H_4_ (~14 kJ mol^−1^). For C_3_H_6_ adsorption, the *Q*_st_ values for MIL-101(Cr) (from 34.3 to 26.7 kJ mol^−1^) and activated carbon (from 38.5 to 32.5 kJ mol^−1^) decrease as the loading increase, while the *Q*_st_ values for HKUST-1, ZIF-8 and Fe_2_Mn-L first gradually decrease from low loading and then increase at high loading (Figure 3d), changing from 48.5, 29.0, and 39.9 kJ mol^−1^, to 35.0, 26.8, and 28.4 kJ mol^−1^, and eventually to 41.0, 35.6, and 34.1 kJ mol^−1^, respectively. *Q*_st_ values of the adsorbents are obviously large than the vaporization enthalpy of C_3_H_6_ (~19 kJ mol^−1^). The rise of *Q*_st_ at high loading probably results from the rising of guest-guest interactions.

The above results revealed HKUST-1 and activated carbon show high affinity to the C2 and C3 olefins, which resulted in their high loading of the olefins even at low-pressure ranges. Relatively, the interactions between the olefins and Fe_2_Mn-L, MIL-101(Cr), or ZIF-8 are low, although, at low pressures, Fe_2_Mn-L and MIL-101(Cr) also show quite high *Q*_st_ values. The interaction between Fe_2_Mn-L and C_3_H_6_ is not strong, but it shows high C_3_H_6_ uptakes, which should be a result of its high porosity and moderate pore size. It is also suggested that the nature of metal ions in the adsorbents does not profoundly affect their C_3_H_6_ and C_2_H_4_ adsorption behaviors. For the adsorptive C_3_H_6_/C_2_H_4_ separation, three key parameters need to be considered, namely, C_3_H_6_/C_2_H_4_ adsorption selectivity, adsorption capacity, and regeneration energy. Based on the above-mentioned results, among the tested adsorbents, HKUST-1 may outperform the others in high adsorption selectivity, while Fe_2_Mn-L may be advantageous in high adsorption capacity and facile regeneration for C_3_H_6_/C_2_H_4_ separation.

The C_3_H_6_/C_2_H_4_ separation performances of other types of adsorbents have also been reported. For example, Han et al. reported two covalent organic frameworks (COFs), CR-COF-1 and CR-COF-2, by one-pot Suzuki coupling and Schiff’s base reaction [5]. The C_3_H_6_ uptakes were 84 and 137 cm^3^ g^−1^, and the C_2_H_4_ uptakes were 38 and 72 cm^3^ g^−1^ at ~1 bar and 298 K for CR-COF-1 and CR-COF-2, respectively. The uptakes are obviously lower than those of Fe_2_Mn-L (291.1 and 87.5 cm^3^ g^−1^). Zhang et al. prepared a blend membrane by doping 15% (mass) polyethylene glycol (PEG600) into the poly(ether-block-amide) (Pebax® 2533) polymer matrix [6]. For a 1:1 binary C_3_H_6_/C_2_H_4_ gas, the Pebax® 2533/PEG600 membrane showed a C_3_H_6_ permeability of 273 barrer and a C_3_H_6_/C_2_H_4_ selectivity of 4.15 at 293 K and 2 bar, and a C_3_H_6_ permeability of 196 barrer and a C_3_H_6_/C_2_H_4_ selectivity of 8.90 at 238 K and 2 bar. The permselectivities are comparable to the C_3_H_6_/C_2_H_4_ IAST selectivity of Fe_2_Mn-L at 298 K and 1 bar for a 1:1 binary C_3_H_6_/C_2_H_4_ gas (7.8).

### 3.3. Dynamic Breakthrough of Binary C_3_H_6_/C_2_H_4_ Gas

To confirm the separation capacities of the adsorbents for a real binary C_3_H_6_/C_2_H_4_ gas, breakthrough experiments were performed for the quartz tubes packed with the adsorbents by using a binary gas of C_3_H_6_/C_2_H_4_ (1:9) at room temperature and ambient pressure. The tested adsorbents all show capability to separate the C_3_H_6_/C_2_H_4_ gas mixture, which is indicated by the gaps between their C_2_H_4_ and C_3_H_6_ breakthrough curves (Figure 4a). Among the five adsorbents, HKUST-1 showed the longest breakthrough time for C_3_H_6_ (530 min g^−1^), indicating a C_3_H_6_ adsorption capacity of 106 cm^3^ g^−1^. The C_2_H_4_ breakthrough time for HKUST-1 was 62 min g^−1^, corresponding to a C_2_H_4_ adsorption capacity of 112 cm^3^ g^−1^. Accordingly, about 842 cm^3^ g^−1^ purified C_2_H_4_ could be obtained in a breakthrough test of HKUST-1. The C_3_H_6_/C_2_H_4_ separation capacity is also high for activated carbon according to its breakthrough curves. It captured 69 cm^3^ g^−1^ C_3_H_6_ and 106 cm^3^ g^−1^ C_2_H_4_, and about 515 cm^3^ g^−1^ purified C_2_H_4_ could be obtained for a breakthrough run. Fe_2_Mn-L captured about 73 cm^3^ g^−1^ C_3_H_6_ before C_3_H_6_ started to penetrate the MOF column. Meanwhile, 153 cm^3^ g^−1^ C_2_H_4_ was adsorbed, and the productivity of purified C_2_H_4_ in each breakthrough run was 504 cm^3^ g^−1^ for Fe_2_Mn-L, slightly less than the productivity of activated carbon. The C_2_H_4_ and C_3_H_6_ adsorption capacities of ZIF-8 and MIL-101(Cr) were obviously lower, and only about 181 and 271 cm^3^ g^−1^ purified C_2_H_4_ could be obtained in each of their breakthrough tests, respectively.

The binary gas breakthrough experiment results are in good accordance with the results of adsorption isotherm measurements for the individual gases, IAST selectivity, and adsorption heat calculations. Specifically, HKUST-1 exhibited high *Q*_st_ and uptakes for both C_3_H_6_ and C_2_H_4_, and its C_3_H_6_/C_2_H_4_ IAST selectivities were also high relative to those of the other tested adsorbents. A high C_3_H_6_/C_2_H_4_ separation performance was indeed observed in the binary gas breakthrough experiment for HKUST-1. Similar results were also observed for activated carbon, except that it exhibited lower adsorption capacities than HKUST-1. For Fe_2_Mn-L, although its C_3_H_6_/C_2_H_4_ IAST selectivities were obviously lower than those of HKUST-1 and activated carbon, it still showed a high C_3_H_6_/C_2_H_4_ separation performance for a real binary gas. Compared with HKUST-1, Fe_2_Mn-L shows lower productivity of purified C_2_H_4_ in a breakthrough run, and the C_2_H_4_ productivities of Fe_2_Mn-L and activated carbon are close. However, it should be noted that the *Q*_st_ values of Fe_2_Mn-L for C_3_H_6_ and C_2_H_4_ adsorption are much lower than those of HKUST-1 and activated carbon, which would lead to its advantage in regeneration with less energy consumption. This conjecture is further sustained by the comparison of C_3_H_6_ and C_2_H_4_ desorption dynamics of the adsorbents. After C_3_H_6_ and C_2_H_4_ were saturatedly adsorbed by the adsorbents during breakthrough experiments, the adsorbents were regenerated by purging a He flow. It was found that the evacuation of adsorbed gases in Fe_2_Mn-L was clearly faster than that of HKUST-1 or activated carbon (Figure 4b).

## 4. Conclusions

In summary, three new isostructural MOFs, Fe_2_M-L (M = Mn^2+^, Co^2+^, or Ni^2+^), have been obtained, which all show large surface areas (BET: ~3100 m^2^ g^−1^) and high pore volumes (~1.3 cm^3^ g^−1^). Adsorption isotherms at room temperature suggest the MOFs uptake ~300 cm^3^ g^−1^ C_3_H_6_ and ~90 cm^3^ g^−1^ C_2_H_4_ at 1 bar. The potential of the MOFs in C_3_H_6_/C_2_H_4_ separation has been further evaluated by IAST selectivity prediction, adsorption heat calculations, and dynamic binary C_3_H_6_/C_2_H_4_ (1:9) gas breakthrough experiments. The predicted IAST C_3_H_6_/C_2_H_4_ adsorption selectivity for Fe_2_Mn-L is ~8 at 298 K and 1 bar. The C_3_H_6_ and C_2_H_4_ adsorption heats for Fe_2_Mn-L are estimated to be 28–40 kJ mol^−1^ and 20–38 kJ mol^−1^, respectively. A binary gas breakthrough experiment confirms the capability of Fe_2_Mn-L to selectively adsorb C_3_H_6_ over C_2_H_4_, producing 469 cm^3^ g^−1^ purified C_2_H_4_ in a breakthrough run. In addition, the C_3_H_6_/C_2_H_4_ separation performances of four other benchmark adsorbents, HKUST-1, MIL-101(Cr), ZIF-8, and activated carbon, were also studied for comparison. The results reveal that Fe_2_M-L are promising adsorbents for C_3_H_6_/C_2_H_4_ separation with low energy penalty in regeneration, although HKUST-1 shows higher C_3_H_6_/C_2_H_4_ adsorption selectivity and productivity of purified C_2_H_4_ than Fe_2_M-L and the other tested adsorbents.

## Figures and Tables

**Figure 1 materials-16-00154-f001:**
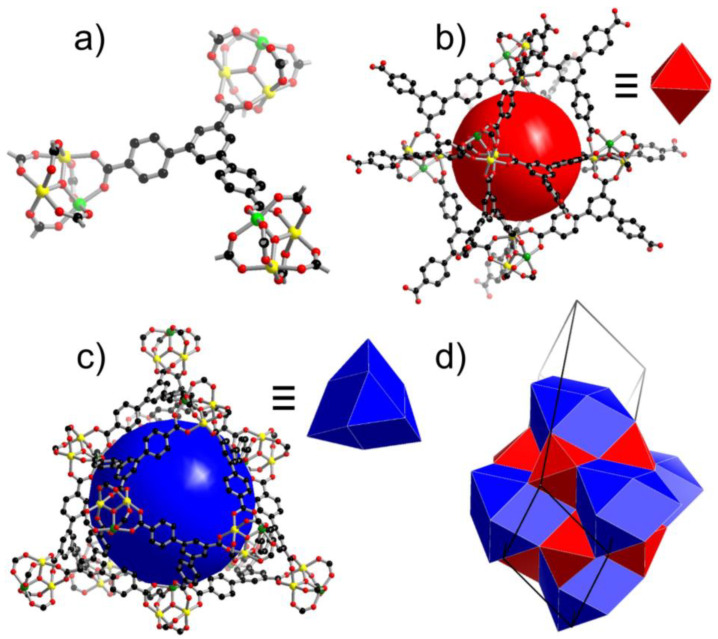
(**a**) two types of building units, (**b**) octahedral, (**c**) tetrakaidecahedral cages, and (**d**) packing of the two types of cages in Fe_2_M-L. Color code: Fe, yellow; M, green; C, black; O, red.

**Figure 2 materials-16-00154-f002:**
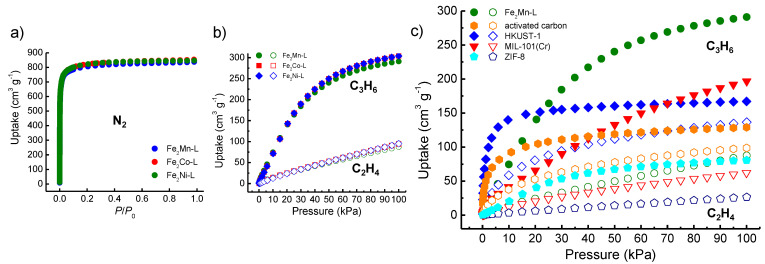
(**a**) Adsorption isotherms of N_2_ measured at 77 K for Fe_2_M-L; (**b**) adsorption isotherms of C_2_H_4_ (open symbols) and C_3_H_6_ (filled symbols) recorded at 298 K for Fe_2_M-L, and (**c**) C_2_H_4_ (open symbols) and C_3_H_6_ (filled symbols) adsorption isotherms at 298 K for HKUST-1, MIL-101(Cr), ZIF-8, activated carbon and Fe_2_Mn-L.

**Figure 3 materials-16-00154-f003:**
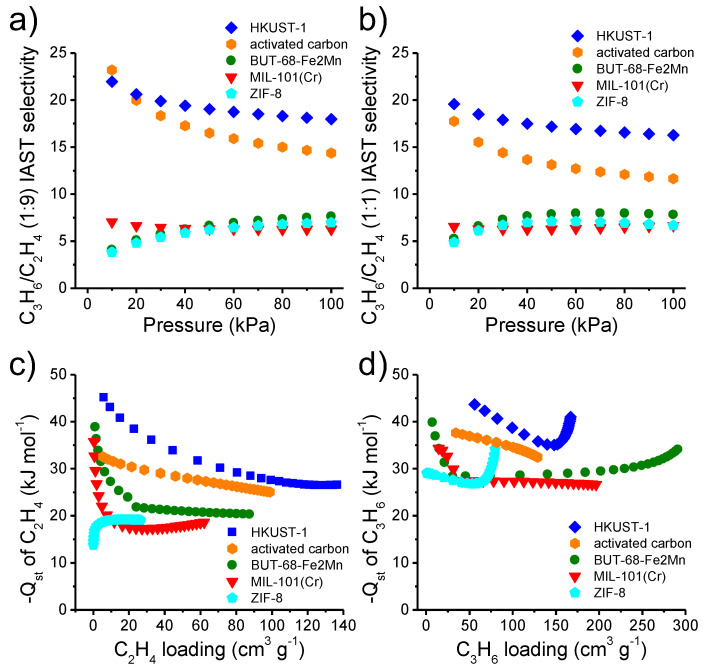
C_3_H_6_/C_2_H_4_ IAST selectivity of the adsorbents for 1:1 (**a**) and 1:9 (**b**) gas mixtures and their isosteric heats of C_2_H_4_ (**c**) and C_3_H_6_ (**d**) adsorption calculated from adsorption isotherms recorded at 273 and 298 K, respectively.

**Figure 4 materials-16-00154-f004:**
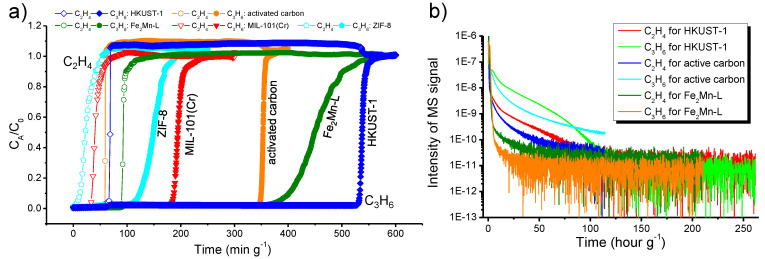
(**a**) Breakthrough curves of 1:9 binary C_3_H_6_/C_2_H_4_ gas in a flow rate of 2 SCCM (standard cubic centimeters per minute) passing through the columns packed with the adsorbents at ambient conditions. Open symbols are for C_2_H_4_, and filled symbols are for C_3_H_6_. C_A_/C_0_: outlet concentration/feed concentration. (**b**) Monitoring the gas desorption processes in the breakthrough columns after adsorption saturation by purging a He flow (10 SCCM).

**Table 1 materials-16-00154-t001:** The porosities of HKUST-1, MIL-101(Cr), ZIF-8, activated carbon, and Fe_2_M-L estimated by their N_2_ adsorption data.

Adsorbent	Pore Volume (cm^3^ g^−1^)	*S*_BET_ (m^2^ g^−1^)	S_Langmuir_ (m^2^ g^−1^)
HKUST-1	0.63	1513.6	1759.3
MIL-101(Cr)	1.47	2691.8	4702.1
ZIF-8	0.85	1510.8	2007.9
activated carbon	0.46	1086.4	1262.6
Fe_2_Mn-L	1.29	3105.2	3599.6
Fe_2_Co-L	1.32	3168.1	3674.5
Fe_2_Ni-L	1.31	3168.9	3674.3

## Data Availability

The data presented in this study are available on request from the corresponding authors.

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
