# Peer review of "Efficient Propylene/Ethylene Separation in Highly Porous Metal–Organic Frameworks"

_materials, 2022, doi:10.3390/ma16010154_

Round 1
Reviewer 1 Report
see the attached file please.
For the crystal structure of Fe2Co-L, the Ñrmax is greater than 1. Usually, it should be less than one. Mention where this peak is located? Is it near the metal center?
1. WR2 values look nice but for the better refinement, these values should be as minimum as possible.
2. For the crystal structure of Fe2Mn-L, the thermal ellipsoids are large for the following atoms, it should be check
C13 1 0.238362 0.238431 0.633351 11.00000 0.15077 0.10658 =
0.08835 -0.05227 -0.07469 0.10592
C12 1 0.232184 0.281207 0.640571 11.00000 0.14257 0.10443 =
0.07058 -0.04390 -0.06049 0.09873
C4 1 0.182219 0.124006 0.599732 11.00000 0.10216 0.10492 =
0.07958 -0.04514 -0.04748 0.08611
C5 1 0.225048 0.170082 0.608577 11.00000 0.12607 0.10406 =
0.08221 -0.05252 -0.06147 0.09896
Also, the same is to be checked for other structures as well.
3. Check the thermal ellipsoids of atoms in other crystal structures as well.
4. In section 2.3. Single-crystal X-ray diffraction, mention the name of software with reference that are used for the graphical representations of single crystal XRD results.
5. Is there any effect on the geometry, cages and packing of cages by changing the metal? Discuss it in section 3.1. Indeed, the Figure 1 is drawn as general case only, but authors have already collected the data for Mn-, Co-, and Ni-based Fe2ML MOFs, therefore it is highly required to fix the Figure 1 for each case separately/independently and then discuses.
6. The introduction section be elaborated from the MOF’s structural point of view of such bimetallic MOFs and their applications.
7. Provide the comparison of these results of separation of C2/C3 busing MOFs with already reported results by different types of adsorbents too.
Reviewer 2 Report
The authors report the synthesis of 3 mixed-metal isostructural MOFs and study their adsorption of C2/C3 hydrocarbons. The results have been compared to some benchmark MOFs and activated carbon. While the manuscript is reasonably suited for consideration, there are some issues which are needed to be addressed prior to that.
1. The main target of the project is presented as separation of propylene and ethylene. However, the introduction does not give a clear description of why that is an important research problem. Also, a discussion of C2-C3 hydrocarbon mixtures with previous literature references should be included.
2. Is the mixed-metal composition 1:1 even for the bulk sample? The manuscript currently does not have any characterisation in this regard.
3. In all the simulated PXRD patterns there seems to be a small peak at ~6.5 degree next to the more intense one (Figure S2). For the all the experimental patterns, that peak is absent. Is there a specific reason for that?
4. All the reported adsorption data can be uploaded as an AIF file, as the accepted practice in the MOF community (https://mof-international.org/adsorption-information-file/).
5. The light hydrocarbon studies have previously been reported for MOFs such as HKUST-1, ZIF-8 and MIL-101(Cr). The corresponding results should be compared and cited wherever possible.
Reviewer 3 Report
In this work, Liu and co-workers study a number of adsorbents for the separation of propylene/ethylene mixtures, including three newly synthesized metal-organic frameworks (MOFs) bearing Fe2M (M = Mn, Co, Ni) clusters as secondary building units. The experimental work, and its description is sound. The new materials were characterized by several methods, and many adsorption studies have been conducted. The work is well-written and presented, it suits the journal and special issue to which it was submitted to, and its results shall be significant for a broad range of materials chemists. Therefore, I recommend the publication of this manuscript in Materials.
The manuscript can be published in its present form. However, I strongly suggest the authors to consider the following minor revisions:
1. Clarify what exactly are ZIF-8, and HKUST-1, especially which metals are present in the structure. Is there any effect of the nature of the metal in the adsorption behavior?
2. Enlarge Figures 2c and 4 to make the data more easily to see. These figures have a large amount of data, and it’s difficult to follow the trends discussed in the text.
3. Although I agree with the reported characterization of Fe2M-L MOFs, the correspondence between the experimental PXRD patterns and simulated ones is not an exact match. Several additional peaks are present in the experimental patterns. The authors should comment on what the additional peaks might be. Have the authors checked whether this could correspond to the pure Fe or pure M MOFs, instead of the mixed metal one?
Round 2
Reviewer 1 Report
Point-by-point responses to the comments are okay.
Reviewer 2 Report
The authors have addressed most questions satisfactorily. The manuscript is now suited for publication after a couple of minor inclusions as below:
- The ICP information (Table S2) should be referred to in the main text.
- The experimental details for this data should be added to the Methods section.
Author Response
General comments: The authors have addressed most questions satisfactorily. The manuscript is now suited for publication after a couple of minor inclusions as below:
Response: Thank you very much for your comments and constructive suggestions. Please find the detailed responses to your comments and suggestions below.
Specific comment 1: The ICP information (Table S2) should be referred to in the main text.
Response: Thank you very much for your suggestion. As suggested, the following sentence has been added into the revised manuscript: “The presence of two types of metal ions with a Fe to M ratio of 2:1 in Fe2M-L was confirmed by ICP-AES measurements for digested samples of the MOFs (Table S2).”
Specific comment 2: The experimental details for this data should be added to the Methods section.
Response: Thank you very much for your suggestion. As suggested, the following sentence has been added into the “Materials and Methods” section of revised manuscript: “Inductively coupled plasma-atomic emission spectroscopy (ICP-AES) measurements were performed with a PerkinElmer Optima 8300 spectrometer.”